# Serotonin signaling mediates protein valuation and aging

**Jennifer Ro[1], Gloria Pak[2], Paige A Malec[3], Yang Lyu[4,5], David B Allison[6], Robert T Kennedy[3], Scott D Pletcher[1,4,5]***

[1]Program in Cellular and Molecular Biology, University of Michigan, Ann Arbor, United States; [2]College of Arts and Science, University of Michigan, Ann Arbor, United States; [3]Department of Chemistry, University of Michigan, Ann Arbor, United States; [4]Department of Molecular and Integrative Physiology, University of Michigan, Ann Arbor, United States; [5]Geriatrics Center, University of Michigan, Ann Arbor, United States; [6]Nutrition Obesity Research Center, University of Alabama at Birmingham, Birmingham, United States

**Abstract** Research into how protein restriction improves organismal health and lengthens lifespan has largely focused on cell-autonomous processes. In certain instances, however, nutrient effects on lifespan are independent of consumption, leading us to test the hypothesis that central, cell non-autonomous processes are important protein restriction regulators. We characterized a transient feeding preference for dietary protein after modest starvation in the fruit fly, *Drosophila melanogaster*, and identified tryptophan hydroxylase (*Trh*), serotonin receptor 2a (5HT2a), and the solute carrier 7-family amino acid transporter, *JhI-21*, as required for this preference through their role in establishing protein value. Disruption of any one of these genes increased lifespan up to 90% independent of food intake suggesting the perceived value of dietary protein is a critical determinant of its effect on lifespan. Evolutionarily conserved neuromodulatory systems that define neural states of nutrient demand and reward are therefore sufficient to control aging and physiology independent of food consumption.

*For correspondence: spletch@umich.edu

**Competing interests:** The authors declare that no competing interests exist.

## Introduction

The availability of dietary protein elicits rapid and significant effects on behavior and lifespan across taxa. Availability of specific nutrients, rather than overall caloric value, may be the driving force for this effect under some circumstances, and dietary protein is particularly important (*Kamata et al., 2014*; *Mair et al., 2005*; *Mayntz et al., 2005*). Protein restriction extends lifespan in crickets (*Reddiex et al., 2013*), flies (*Mair et al., 2005*), mice (*Solon-Biet et al., 2014*), and probably humans (*Levine et al., 2014*). Nearly all research into the mechanisms of this phenomenon has focused on the consequences of amino acid imbalance within cells, mostly through investigation of the TOR pathway and its effectors (*Efeyan et al., 2015*). Remarkably, however, many of the effects of diet manifest independently of food consumption, likely through global integration of nutrient signals and cell non-autonomous responses to those signals directed by the nervous systems (*Linford et al., 2011*; *Mair et al., 2005*; *Taylor et al., 2014*). Indeed, sensory neurons in *Drosophila melanogaster* and *Caenorhabditis elegans* can promote or limit lifespan depending on the specific neurons involved (*Alcedo and Kenyon, 2004*; *Apfeld and Kenyon, 1999*; *Libert et al., 2007*), and the first instance of sensory modulation of lifespan in mice was recently reported (*Riera et al., 2014*).

Instances in which animals adjust their behavior to emphasize intake of specific nutrients are well-known both in the wild and in the laboratory settings. For example, predatory spiders are known to select their prey depending on predicted nutrient composition (*Mayntz et al., 2005*), spider

**eLife digest** Limiting the amount of protein eaten, while still eating enough to avoid starving, has an unexpected effect: it can slow down aging and extend the lifespan in many animals from flies to mice. Previous work suggests that how an animal perceives food can also influence how fast the animal ages. For example, both flies and worms actually have shorter lifespans if their food intake is reduced when they can still "smell" food in their environment. However, the sensory cues that trigger changes in lifespan and the molecular mechanisms behind these effects are largely unknown.

Ro et al. therefore asked whether fruit flies recognize protein in their food, and if so, whether such a recognition system would influence how the flies age. Flies that had been deprived of food for a brief period tended to eat more protein than other flies that had not been starved. Ro et al. then revealed that serotonin, a brain chemical that can alter the activity of nerve cells, plays a key role in how fruit flies decide to feed specifically on foods that contain protein. Further experiments revealed also that flies age faster when they are allowed to interact with protein in their diet independently from other nutrients, despite eating the same amount. Disrupting any of several components involved in serotonin signaling protected the flies from this effect and led to them living almost twice as long under these conditions.

Ro et al. propose that the components of the recognition system work together to determine the reward associated with consuming protein by enhancing how much an animal values the protein in its food. As such, it is this protein reward or value – rather than just eating protein itself – that influences how quickly the fly ages. Further work is now needed to understand how the brain mechanisms that allow animals to perceive and evaluate food act to control lifespan and aging.

monkeys tightly regulate daily protein intake in the wild (*Felton et al., 2009*), and laboratory mice balance their macronutrient intake differently under influence of a drug (*Shor-Posner et al., 1986*). Changes in behavior of this sort require a dynamic process of context-dependent valuation of nutrients, which almost surely includes an integration of sensory perception of ecological availability and an internal assessment of nutrient demand. State-dependent valuation and how it drives behavior have been studied in both invertebrates and vertebrates (*Pompilio et al., 2006*; *Tindell et al., 2006*). Context-dependent value of sugars has been established for oviposition preference in *Drosophila* (*Yang et al., 2008*), and it is likely that food preference behavior also includes a similar context-dependent signaling process (*Ribeiro and Dickson, 2010*).

Unfortunately, the molecular mechanisms underlying how animals determine the value of certain nutrients in a context-dependent manner are not well understood. Previous studies have sought an understanding of the neural bases for assessing protein and carbohydrate availability (*Thibault and Booth, 1999*) because, phenotypically, these two macronutrients influence many biological activities, including fat accumulation, reproductive behavior, and lifespan (*Lee et al., 2008*; *Skorupa et al., 2008*; *Tatar et al., 2014*). For a small fraction of these phenotypes, we have some understanding of mechanism. Genetic and neuronal manipulations have identified the biogenic amine, dopamine, as important for oviposition preference for dietary sugars in *Drosophila* (*Yang et al., 2015*) and for recognition of the nutritive quality of sugar in mice (*de Araujo et al., 2008*). A second biogenic amine, serotonin, has been implicated as an indicator of carbohydrate satiety and, less clearly, for influencing protein or lipid feeding. However, these studies are less well-defined and pharmacological approaches have been used. (*Johnston, 1992*; *LeBlanc and Thibault, 2003*; *Leibowitz and Alexander, 1998*; *Leibowitz et al., 1993*; *Magalhães et al., 2010*). Unlike other biogenic amines serotonin is produced in the brain as well as in peripheral tissues. In many organisms the majority of serotonin is produced in the gastrointestinal track (*Gershon et al., 1965*), using a distinct synthetic pathway (*Neckameyer et al., 2007*). Pharmacologic manipulation is, therefore, not sufficient to distinguish peripheral effects from those on central processing, such as satiety, reward, and overall nutrient value.

We postulated that central mechanisms in the brain that drive cell non-autonomous responses to protein valuation might be important determinants of aging. As described above, there is evidence that organisms forage to balance their intake of specific nutrients rather than merely to meet

energetic requirements, and even humans are known to make feeding decisions based on dietary protein (*Griffioen-Roose et al., 2011*). Although the molecular mechanisms for such choices are not well understood, important components of the process must include the ability to sense protein, to assess the value of protein relative to demand, and to execute behavioral and physiological responses that maintain protein homeostasis. We therefore initially sought insight into mechanisms of short-term behavioral choice in response to protein manipulations, with the expectation that targeting specific components of this mechanism might influence aging through valuation itself, independent of feeding or total nutrient intake. Here, we establish that serotonin signaling in the CNS through one serotonin receptor, receptor 2a, is required for protein preference by determining the value of protein at the time of physiological demand. We also provide the first documented functional connection between amino acid transporter, JhI-21, and serotonin signaling in the context of macronutrient selection. We further demonstrate that modulators of protein value also mediate diet-dependent lifespan when animals are exposed to a complex nutrient environment where they are presumably required to continuously evaluate internal nutritional state relative to the availability of individual nutrients in the environment. These results highlight how the macronutrient valuation process itself, in the context of perceived availability and demand, can influence the aging process independent of food consumption.

## Results

### Drosophila develop a preference for protein under mild starvation

We hypothesized that mechanisms underlying behavioral responses to protein availability would also be important determinants of lifespan and therefore sought first to identify central mechanisms involved in protein-dependent feeding decisions. We characterized a dynamic and transient protein-seeking behavior in *Drosophila* in response to nutrient demand. Using a new real-time feeding monitoring system called FLIC, which quantifies all contact interactions an individual fly has with food (*Ro et al., 2014*; http://www.wikiflic.com), we found that after mild starvation both male and female flies preferred a sugar diet supplemented with autolyzed yeast (the major protein source in the fly diet) over a diet composed exclusively of sugar (*Figure 1A*, 24 hr starvation). This preference persisted for just over one day, with longer food deprivation increasing protein preference (*Figure 1A*; *Figure 1—figure supplement 1A*). On the other hand, fully-fed animals consumed a sugar-only diet more often throughout the experiment (*Figure 1A*, fully-fed). Food choice was not based on caloric content because starved flies also chose a 1% protein diet over an isocaloric 1% sugar diet and an increase in protein concentration to a hypercaloric 5% did not affect this preference (*Figure 1B*; *Figure 1—figure supplement 1B*). The FLIC system identifies every interaction with each food type and quantifies when this occurs and for how long. Using this information we found that total feeding time was a much stronger predictor of total protein feeding time than it was of sugar feeding time (*Figure 1—figure supplement 1C*). Moreover, flies with more total feeding time were more likely to choose a protein containing food as their first meal and to have a stronger protein preference over the course of the experiment (*Figure 1C*). These results indicate that individual flies are affected differently by 24 hr starvation and that those under high nutrient demand (identified by large total feeding time) tend to choose protein as their first meal. These flies also consume greater amounts of protein over the course of the experiment, presumably to ameliorate their deficit. Following mild starvation, flies preferred autolyzed yeast, which is primarily composed of short peptide/single amino acids (*Figure 1—figure supplement 1D*), to the same extent as a pure complex protein such as bovine serum albumin (BSA) (*Figure 1D*) over a wide range of concentrations, suggesting that behavioral preference for yeast is driven in whole or in part by its protein content. Twenty-four hour starvation had no detectable effect on internal protein levels (*Figure 1E*), suggesting that preference was independent of gross protein stores. Protein preference is, therefore, not limited to reproductively active female flies (*Ribeiro and Dickson, 2010*; *Vargas et al., 2010*) and is more dynamic than previously suspected.

### Serotonin signaling through 5HT2a mediates protein preference

To identify mechanisms underlying protein-feeding preference, we performed a candidate screen designed to disrupt putative nutrient sensing pathways, sensory systems, and reward circuits

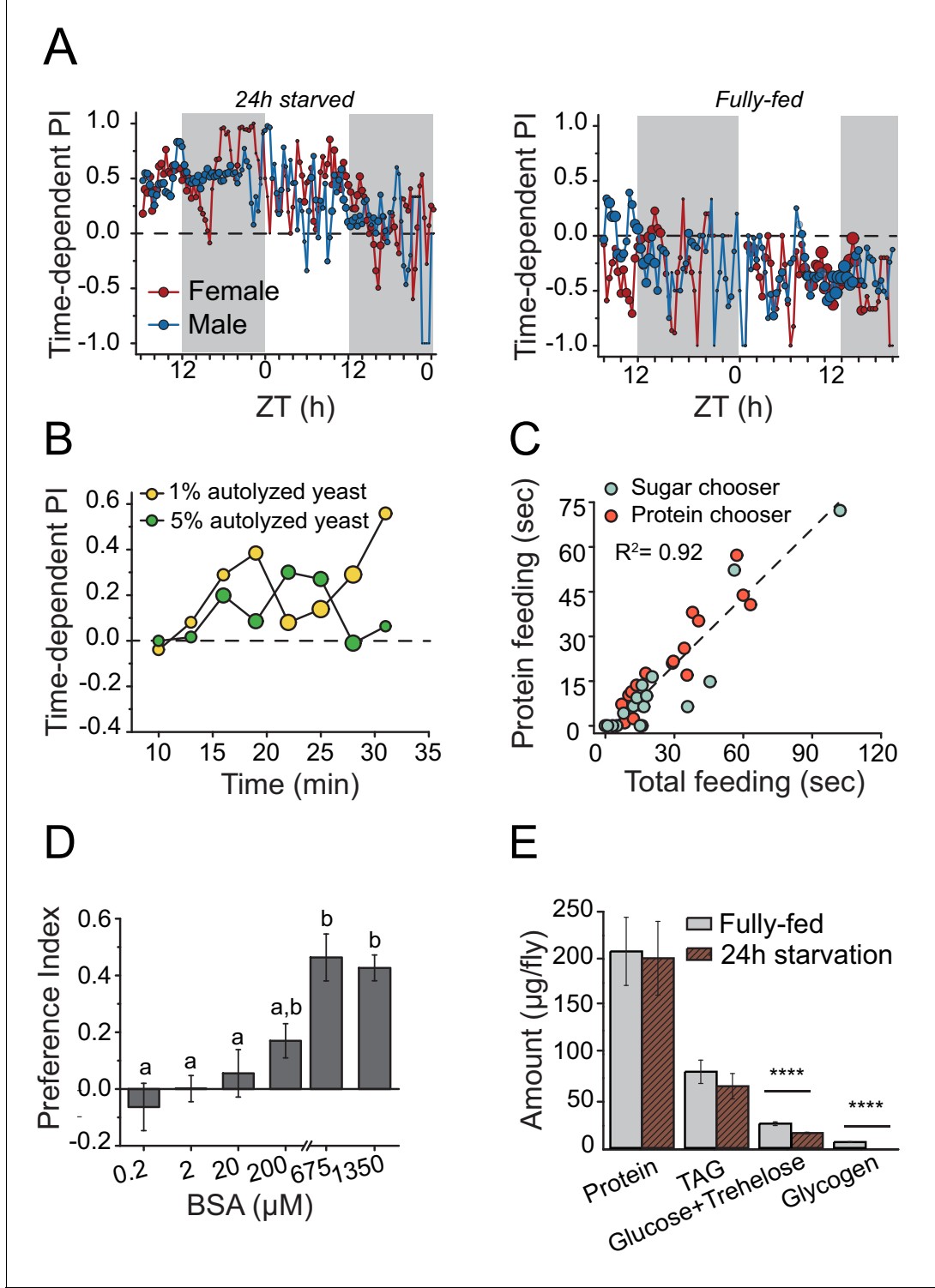

**Figure 1.** Drosophila demonstrates energy-state dependent protein feeding preference. (**A**) Male and female *Canton-S* flies' real-time feeding preference over 24 hr. The choice was given as 2% autolyzed yeast (w/v) +1% sucrose (w/v) vs. 1% sucrose (w/v). Gray shades on the graphs indicate "light-off" periods. The size of the symbols is proportional to the number of flies that were feeding during the given time period (fully-fed female flies N=17; fully-fed male flies N=21; starved female flies N=10; starved male flies N=11). A Preference Index (PI) = 1 indicates complete preference for the yeast-containing food. (**B**) Time-dependent PI plot from flies given a choice between 1% sucrose vs. 1% autolyzed yeast or 1% sucrose vs. 5% autolyzed yeast. Similar yeast preference is observed in both experiments, and in the first experiment the diets are isocaloric. (**C**) Flies that had increased total food consumption during the choice assay were likely to eat more protein meal. From the continuous FLIC data, we identified, for each fly, its first meal

*Figure 1 continued on next page*

*Figure 1 continued*

choice (sugar or protein chooser), time spent on protein feeding, and total feeding time at the end of a 30 min choice experiment. Flies were given a choice between isocaloric 1% sucrose vs 1% autolyzed yeast. Linear regression analysis revealed that total feeding time positively correlated with the protein feeding time (F(1,32)= 173.7, P<1.8E−14). (D) 24 hr-starved female flies' BSA preference was dose-dependent. Bars indicate the mean and the standard error of the mean (SEM). (N= 8–14 per each concentration treatment. Letters differentiate groups that are significantly different from one another as determined by Tukey's multiple-comparison at α=0.05) (E) Quantification of stored nutrient levels in fully-fed or 24 hr-starved female flies. Flies lost a significant amount of carbohydrate reserves after 24 hr of starvation. (P values determined by two-way ANOVA, followed by Tukey's multiple-comparison test. ***P≤0.0001).

The following figure supplement is available for figure 1:

**Figure supplement 1.** Characteristics of protein feeding behavior in flies.

---

(*Figure 2A*; *Supplementary file 1*). Among all manipulations, disruption in serotonin signaling repeatedly reduced protein preference in starved flies (*Figure 2A* blue bars; *Supplementary file 1* candidates 63, 65, 67, and 68). Silencing of serotonergic neurons and treatment of flies with a serotonin receptor 2a (*5HT2a*) antagonist, ketanserin (*Colas et al., 1995*), eliminated preference entirely, such that starved animals exhibited an aversion to dietary protein that mimicked what we routinely observed from fully-fed, control animals.

We verified a role for serotonin signaling in protein feeding decisions using additional genetic manipulations. Flies with significantly reduced tryptophan hydroxylase (*Trh*), a rate-limiting enzyme for neuronal serotonin synthesis (*Neckameyer et al., 2007*) that is the *Drosophila* homologue of TPH2, exhibited a reduction in preference for both autolyzed yeast and BSA compared with control animals (*Figure 2B,C*; *Figure 2—figure supplement 1A,B*; *Figure 2—figure supplement 2A*). In *Drosophila*, there are five known serotonin receptors, each of which is evolutionarily conserved: *5HT1a, 5HT1b, 5HT2a, 5HT2b, and 5HT7* (D. E. Nichols and C. D. *Nichols, 2008*). We verified a role for *5HT2a* by examining flies that carried two independent *5HT2a* mutant alleles (*Figure 2D*, *Figure 2—figure supplement 1C,D*; *Figure 2—figure supplement 2B*), flies transheterozygous for each allele (*Figure 2—figure supplement 1E*), as well as flies with RNAi-mediated knock-down of *5HT2a* transcript (*Figure 2—figure supplement 1F*). All of the *5HT2a* manipulations abrogated the preference for protein following mild starvation, effectively recapitulating *Trh* mutation. Loss of function in three other serotonin receptors (*5HT1a, 5HT1b, and 5HT2b*; 5HT7 mutant animals were not viable) had no effect on the preference phenotype (*Figure 2—figure supplement 3A*). Disruption of serotonin signaling did not disrupt other feeding preferences because *Trh* and *5HT2a* mutant flies exhibited expected choice behaviors when presented with sweet vs bitter tastes or sweet vs sweeter food (*Figure 2—figure supplement 3B*). Furthermore, when fully fed, the feeding behaviors of *Trh* and *5HT2a* mutant flies are not different from control animals (*Figure 2—figure supplement 3C*).

We next sought to better understand the temporal dynamics through which serotonin regulates protein preference. To determine whether serotonin signaling is required during starvation and/or during food choice, we selectively inactivated serotonergic neurons (*Trh-GAL4>UAS-shi^{ts}*) during specific periods of the experiment. We found that flies retained their preference for dietary protein when serotonergic neurons were inactivated only during the starvation phase. Inactivation only during the choice phase, however, was sufficient to abolish preference and phenocopy the *Trh* mutant, implying that serotonin may be involved specifically in protein reward after starvation (*Figure 3A*). Inhibition of serotonin signaling did not alter total food intake during the choice test, demonstrating that the lack of protein preference seen in transgenic flies is not an artifact of reduced feeding (*Figure 3—figure supplement 1A*).

If serotonin is involved in post-ingestive reward, then protein intake, but not starvation, would be expected to increase serotonin levels in the CNS. Indeed, we observed that serotonin levels in the head were unchanged after starvation but increased 100–200% after flies consumed protein compared with the fully-fed condition (*Figure 3B*). Because transcriptional regulation of serotonin receptors can directly affect behavioral output (*Albert and François, 2010*), we examined whether mRNA levels of serotonin receptors were altered in the head of the fly following protein ingestion. Much like the temporal profile of serotonin concentration, we observed an acute increase in the abundance of *5HT2a* mRNA, but not transcript from other serotonin receptors, in the heads of flies that

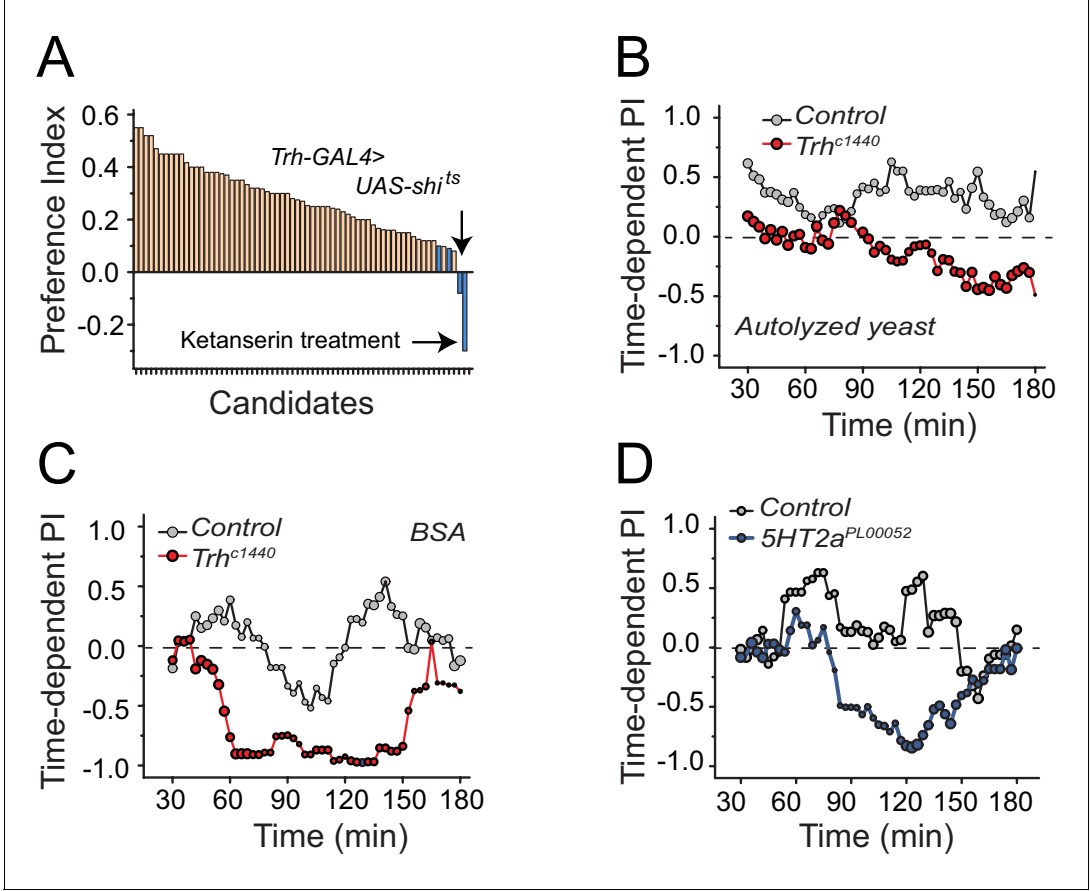

**Figure 2.** Serotonin signaling through receptor 2a modulates protein preference. (**A**) Summary results from a candidate reverse genetic screen. We found that disruption of serotonin signaling consistently abrogates protein preference (See **Supplementary file 1** for listed candidates). We used BSA as the protein source. Blue bars indicate manipulations of serotonin signaling that strongly disrupted protein preference. (**B-C**) Time-dependent PI plot from 24 hr-starved *Trh* mutant and control flies given a choice between sucrose-only or sucrose plus autolyzed yeast or BSA. When the protein source was autolyzed yeast, the cumulative preference index for Canton S files was $0.43 \pm 0.1$ (Mean ± SEM) whereas *Trh* mutant flies was $-0.18 \pm 0.06$ (Student's t-test; $P \leq 0.001$). When the protein source was BSA, cumulative PI for Canton S. files was $0 \pm 0.1$ whereas *Trh* mutant flies was $-0.43 \pm 0.2$ (Student's t-test; $P \leq 0.05$). (**D**) Time-dependent PI plot from 24 hr-starved flies with *5HT2a^PL00052* mutant allele and *Canton S.*, control flies given a choice between sucrose-only or sucrose plus BSA. Cumulative PI for the control was $0.2 \pm 0.2$ and for the mutant was $-0.21 \pm 0.2$ (Student's t-test).

The following figure supplements are available for figure 2:

**Figure supplement 1.** The effect of serotonin manipulations on protein choice behavior is robust and reproducible.

**Figure supplement 2.** Serotonin signaling influences food early food choice.

**Figure supplement 3.** Serotonin receptor 2a is required for protein preference and mutation in Trh or 5HT2a does not affect other taste modality-dependent choice behavior.

were starved of all nutrients and refed with protein for 3 hr (*Figure 3C*). *5HT2a* mRNA abundance was not affected by starvation alone, and it returned to the level of fully-fed animals after 24 hr of protein feeding, which is coincident with their loss of feeding preference (e.g., *Figure 1A*). Together, these results suggest that protein reward and preference are established in a relatively short time (<3 hr) and regulated by serotonin signaling through 5HT2a.

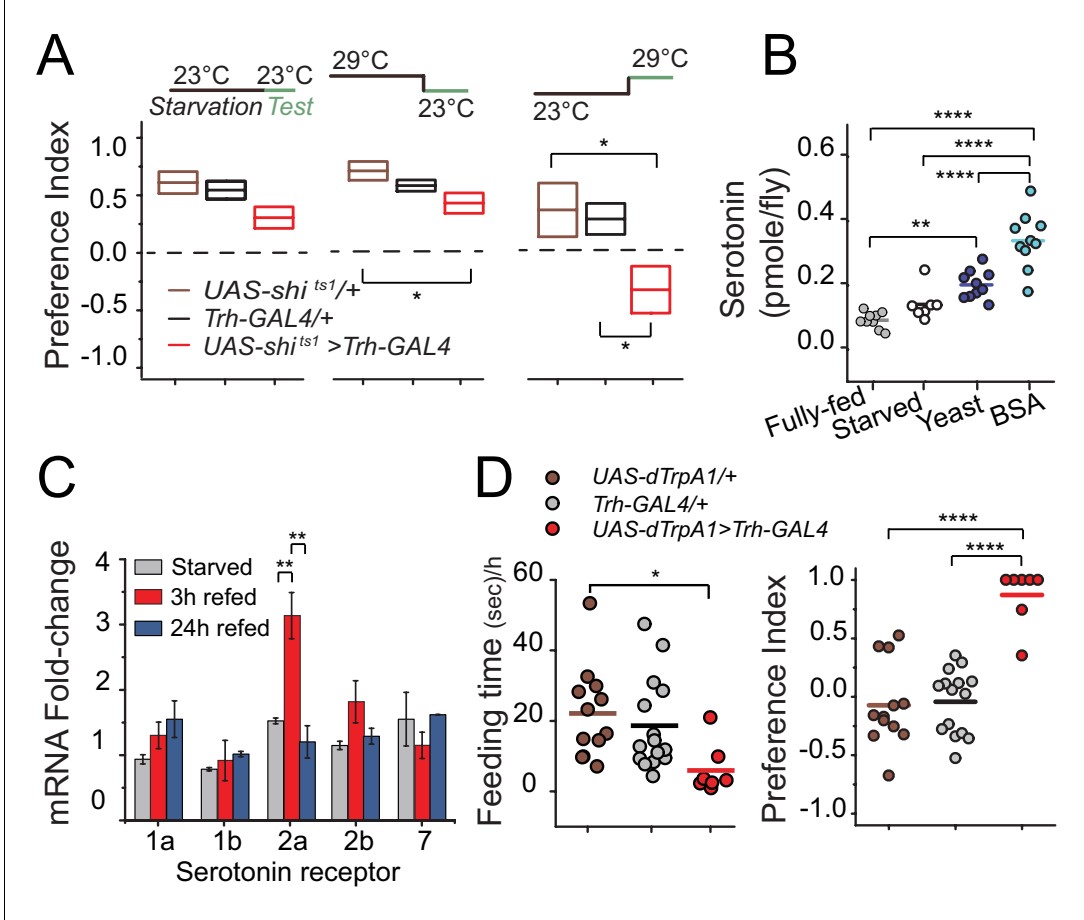

**Figure 3.** Temporal dynamics of serotonin signaling during protein choice behavior. (A) Serotonin signaling is required during the choice test to develop protein preference. Flies were placed in either 23°C or 29°C during starvation or choice test as noted in the diagram above each plot. The box plots indicate mean and SEM. Statistical significance for genotype effect within each temperature-shift experiment was determined using one-way ANOVA followed by Fisher's multiple-comparison, N=8–11/genotype (*P≤0.05). (B) Serotonin abundance in the heads of flies following specified diet treatment. Serotonin significantly increased when animals were allowed to refeed on autolyzed yeast or BSA for three hours after 24 hr starvation. Individual symbols represent measures based on 5 female fly heads, and lines denote the mean. Statistical significance for diet effect was determined using one-way ANOVA followed by Tukey's multiple-comparison; N=7–10/treatment (**P≤0.01, ****P≤0.0001). (C) Neuronal mRNA abundance of five serotonin receptors. Abundance of *5HT2a* transcript acutely increased during 3 hr protein refeeding after starvation. Statistical significance for the treatment effect was determined using one-way ANOVA followed by Tukey's multiple-comparison (**P≤0.01). (D) Hyper-activation of serotonergic neurons is sufficient to suppress feeding behavior (left) and induce protein preference even in the absence of starvation (right). We observed feeding behaviors in 7 out of 14 flies during the choice test. Individual symbols indicate measures from single flies, and lines denote the mean value among biological replicates. Flies were fed standard 10% sugar/yeast medium prior to testing. Statistical significance for the genotype effect was determined using one-way ANOVA followed by Tukey's multiple-comparison (*P≤0.05, ****P≤0.0001).

The following figure supplement is available for figure 3:

**Figure supplement 1.** Effect of neuronal inhibition or activation of central serotonergic neurons on feeding behavior.

## Serotonin signaling through 5HT2a establishes the value of dietary protein

We envisioned at least two ways that serotonin might influence protein reward. First, it may be involved in transducing sensory perception of protein. Second, it may be important for higher-order processing of the value of ingested protein. To distinguish these hypotheses, we hyper-activated serotonergic neurons in fully-fed flies using targeted expression of a heat-sensitive Trp channel (*Trh-GAL4>UAS-dTrpA1*). If serotonin acts in protein sensing, both foods would be interpreted equally as containing protein, and thus we would expect flies to lack preference in our choice test regardless of

their starvation state. In contrast, if serotonin acts to increase the value of consumed protein, we would predict that hyper-activation of serotonergic signaling would add value to protein meals and reinforce protein feeding in the absence of starvation. Although activation of serotonergic neurons suppressed feeding as we expected (*Gasque et al., 2013*) (*Figure 3D* left), when fully-fed flies did eat, they showed a near absolute preference for protein-containing food (*Figure 3D* right; *Figure 3—figure supplement 1B*). These results support the notion that serotonergic signaling increases the value of dietary protein in an energy-state dependent manner.

## The perceived value of dietary protein modulates lifespan via serotonin signaling

We hypothesized that serotonin's role in ascribing value to ingested protein may be important for lifespan, given the noted importance of this nutrient in aging (*Gallinetti et al., 2012*; *Mair et al., 2005*). Laboratory protocols that are standard in the aging field employ fly diets that consist of sugar and brewer's yeast, the sole source of dietary protein, combined in an agar medium in fixed ratios. On a standard laboratory diet of low or intermediate yeast content, *Trh* mutants were long-lived, and loss of *5HT2a* did not affect lifespan (*Figure 4—figure supplement 1A,B*). If serotonin influences aging primarily through the physiological changes in response to the amount of protein consumed, then loss of *Trh* or *5HT2a* would be expected to reduce or eliminate diet-dependent changes in lifespan in fixed-diet conditions (*Skorupa et al., 2008*). We found that *Trh* mutant flies responded similarly to control animals, while *5HT2a* mutants showed a reduced response, when diet was manipulated in this traditional manner (*Figure 4—figure supplement 1A,B*).

While conventional fixed diets have been used effectively to examine physiology in response to total nutrient availability, we reasoned they would obfuscate serotonin's regulatory effects on aging because the animals would not be able to respond to nutrients individually. Flies on a fixed diet can regulate meal size but not individual nutrient intake, resulting in the possibility that feeding behaviors may be dominated by one nutrient at the expense of the other. For example, if the fixed mixture contains a high concentration of a certain nutrient, flies may cease feeding prematurely to prevent overeating that nutrient even if the demand for another is yet to be met. We therefore aged flies in more complex dietary environments where they can freely choose between two distinct food sources. In three control diets flies were provided with either a 10% sucrose-only, 10% yeast-only, or a fixed 10% sucrose/10% yeast diet on both sides of a divider. For the choice environment, flies were provided with 10% sucrose on one side of the divider and 10% yeast on the other, allowing them to freely interact with individual macronutrients (*Figure 4—figure supplement 2A*).

We observed striking effects of the choice environment on lifespan and physiology. As expected, control flies were shortest-lived in a sucrose-only diet (*Lee et al., 2008*; *Skorupa et al., 2008*). On the other hand, we were surprised to observe that control animals in the choice environment lived substantially shorter than animals maintained in an isocaloric fixed diet or in one consisting of yeast-only (*Figure 4A*) despite exhibiting equivalent health metrics when young (e.g., *Figure 4—figure supplement 2B*). Consistent with our observations using standard protocols, *Trh* mutant flies lived modestly longer than control flies on all control diets (*Figure 4A*; Sucrose-only, Yeast-only, and Fixed diet), while *5HT2a* mutants did not. However, when the flies were aged in the choice environment we observed a near doubling (90% increase) of mean lifespan of both *Trh* and *5HT2a* mutant animals (*Figure 4A*, Choice diet).

The dietary choice paradigm influences feeding behaviors but does so equivalently for control and mutant animals. Using an indigestible dye mixed in both food wells we found that all flies consumed more food in the choice environment compared with an isocaloric fixed diet (*Figure 4B*, left two bar groups). By labeling foods in individual wells we found that most of that increased consumption was due to a boost in sugar feeding (*Figure 4B*, right two bar groups). The increase in total feeding was unexpected, but the compositional intake is consistent with previous reports (*Lee et al., 2008*; *Maklakov et al., 2008*) and with our data (*Figure 1A*) showing that fed flies, similar to other insects, choose to consume proportionally more carbohydrate than protein. Fixed diets of this composition are associated with high reproductive output and a reduced lifespan (*de Araujo et al., 2008*; *Skorupa et al., 2008*).

One possible explanation for the exceptional longevity of *Trh* and *5HT2a* mutant flies in the complex nutrient environment is behavioral protein restriction or reduction in total food consumption. However, if mutant animals were simply eating less, we would expect to observe a similar degree of

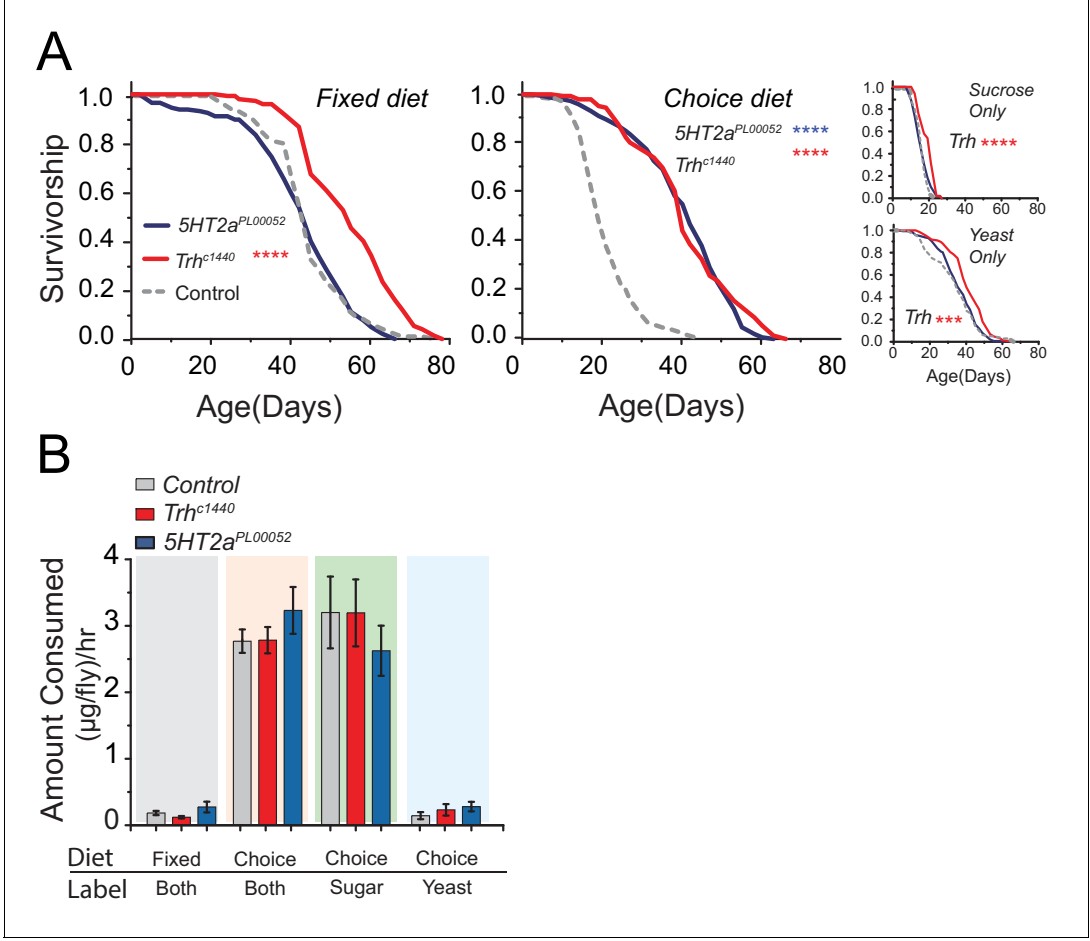

**Figure 4.** Serotonin modulates lifespan. (**A**) *Trh* mutants live significantly longer than control flies when aged on the fixed diet (log-rank test), as well as on the choice diet (long-rank test). *5HT2a* mutants live significantly longer than control flies on the choice diet (log-rank test). *Trh* mutants also live significantly longer in the sucrose-only and yeast-only diets. (***P≤0.001, ****P≤0.0001) (**B**) The amounts of total food (both fixed and choice diets) and individual nutrients (choice diet only) consumed by *Trh* and *5HT2a* mutant flies were statistically indistinguishable from control flies regardless of the diet environment (two-way ANOVA, all P>0.05).

The following figure supplements are available for figure 4:

**Figure supplement 1.** Lifespan analysis of *Trh* and *5HT2a* mutants in the conventional protein restriction diet.

**Figure supplement 2.** Health and Lifespan of *Trh* and *5HT2a* mutants in various diet conditions.

life extension in all diets, which we did not (e.g., *Figure 4A*). More importantly, we found no differences among genotypes in total food consumption in any condition, and in the choice environment, where lifespan extension was greatest, mutant flies consumed a sugar:protein ratio that was statistically indistinguishable from control animals (*Figure 4B* right two bar groups). We therefore find no evidence that serotonergic modulation of lifespan is due to self-induced diet-restriction.

A second possible explanation is that increased carbohydrate consumption alone causes shorter lifespan and that disruption of serotonin signaling protects animals from this effect. Several lines of evidence make it clear that this is not the case. First, mutant flies show the same extent of increased sugar consumption as control animals in the choice environment (*Figure 4B*, third bar group). Second, the lifespans of *Trh* and *5HT2a* mutant flies are reduced to a similar extent as control animals when the carbohydrate content of a fixed diet is increased (*Figure 4—figure supplement 2C,D*), showing that their lifespan is equally or more sensitive to dietary carbohydrate. Third, gross metabolites that are known to be influenced by food intake, including total protein, fat, and

carbohydrate abundances (*Lee et al., 2008*; *Skorupa et al., 2008*), are strongly increased in the choice environment, and mutant and control animals are similarly affected (*Figure 5*). It appears, therefore, that *Trh* and *5HT2a* mutant flies are eating the same amount of the same food, and they are processing nutrients similarly to control animals, in both the fixed- and choice-diet regimes. Yet lifespan of the mutant flies is dramatically longer in the complex dietary environment. Based on the roles of serotonin and 5HT2a in the valuation of dietary protein, we propose that the macronutrient valuation process itself is a potent factor that modulates organismal aging independent of food consumption.

## An amino acid transporter, Jhl-21, is required for serotonergic valuation of protein

The cellular and metabolic processes upstream of serotonin that are required for protein valuation remain to be determined. Sensory perception is likely to be important, but at present, specific amino acid taste receptors have not been identified in *Drosophila*, although this is an active area of research. Our examination of chemosensory receptors known to affect lifespan failed to reveal effects on protein choice behavior (*Supplementary file 1*). Furthermore, manipulation of canonical intracellular amino acid sensing, such as RNAi-mediated knocked-down of GCN2 in starved animals (*Figure 2A*, *Supplementary file 1* #23) or suppression of TOR signaling through rapamycin or over-expression of dominant negative RagA in fully-fed animals, had no effect on protein preference (*Figure 6—figure supplement 1A,B*).

Surprised by the inability of canonical TOR regulators to impact protein preference, we turned our attention to amino acid transporters as possible upstream modulators of serotonin signaling in the context of protein reward. Amino acid transporters may act as general sensors, as well as carriers, of nutrients (*Hyde et al., 2003*; *Nicklin et al., 2009*). Solute carrier 7 protein family members are candidates for such a role, and the leucine transporter SLC7A5 is required to initiate amino acid dependent activation of TOR signaling (*Taylor, 2014*; *Verrey et al., 2004*). A BLAST analysis comparing vertebrate SLC7A5 to the fly protein database revealed the gene *juvenile hormone inducible 21 (Jhl-21)* as the most likely homolog (see also, *Piyankarage et al., 2010*). Interestingly, we found that mutation in *Jhl-21* abolished protein preference in starved flies (*Figure 6A*; *Figure 6—figure supplement 1C,D*). We also discovered evidence that JhI-21 acts upstream of serotonin signaling; *JhI-21* mutant animals failed to increase neuronal serotonin after 3 hr of yeast feeding, establishing that it is required for protein-dependent serotonin production (*Figure 6B*). *JhI-21* mutant females

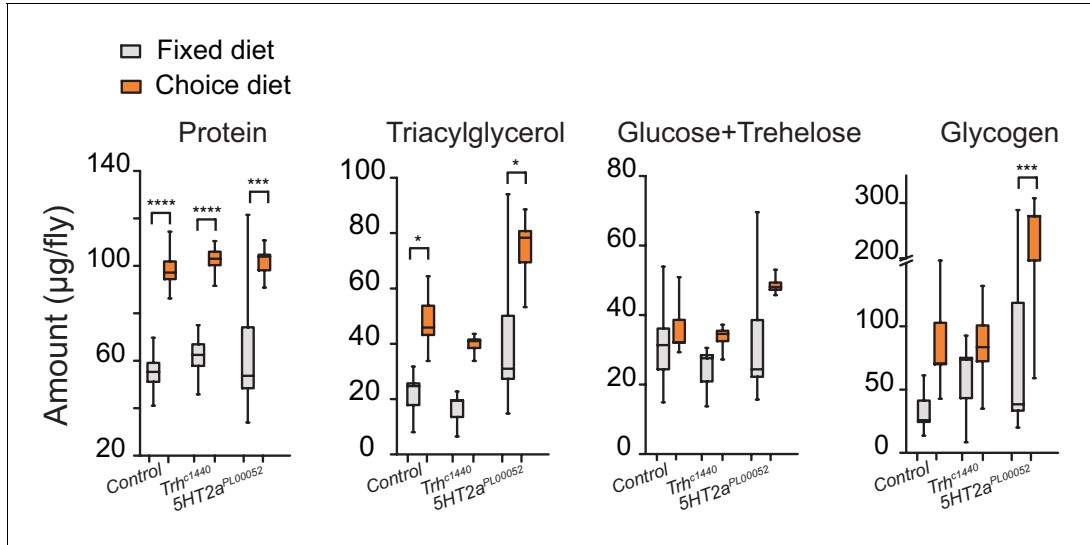

**Figure 5.** Stored nutrient levels in flies kept in fixed or choice diets. Box indicates median and 1 SEM with whiskers showing 10–90% quantile. Significance of the diet effect within the genotype per metabolite was determined by Tukey's multiple-comparison after two-way ANOVA (*P≤0.05, ***P≤0.001, ****P≤0.0001).

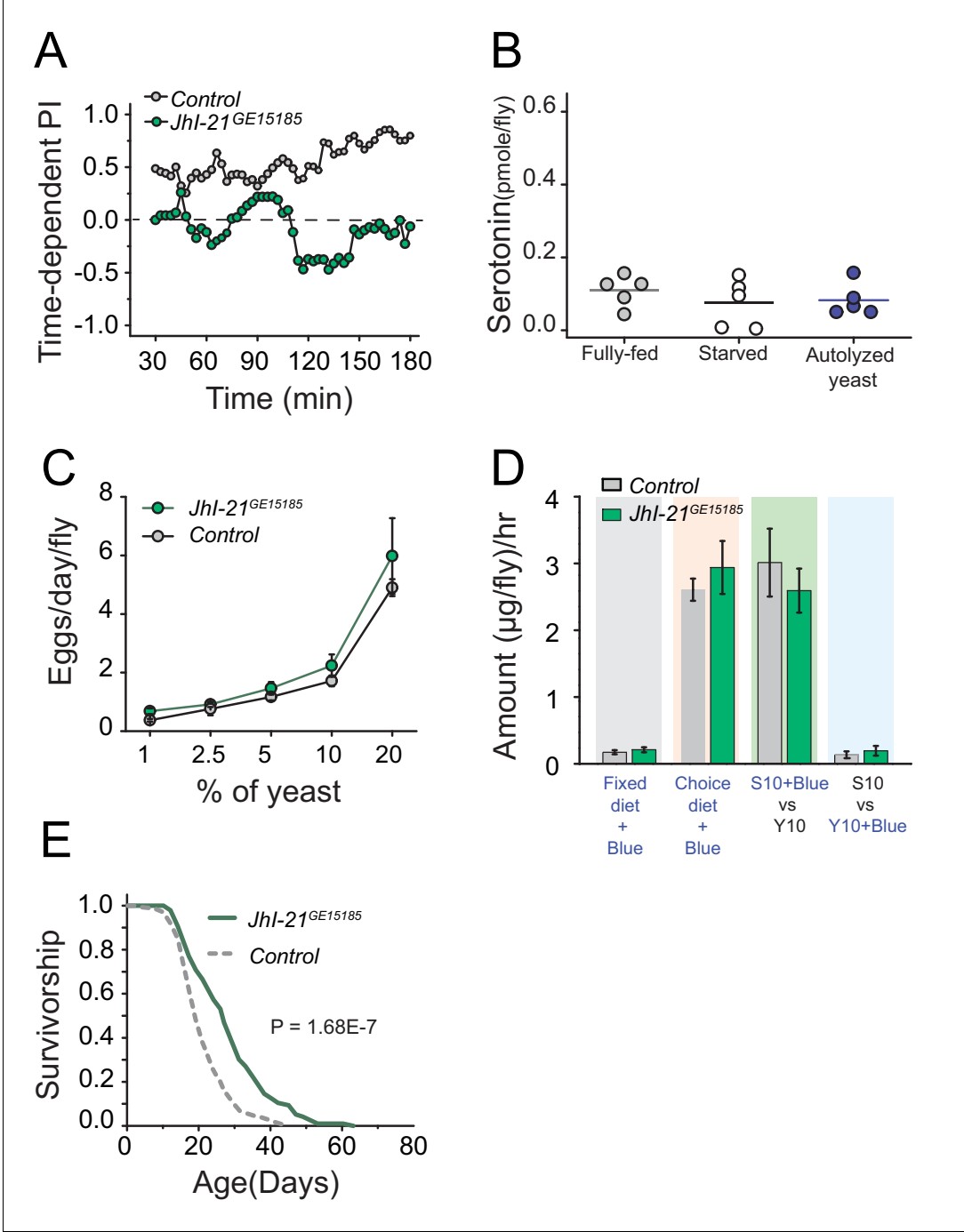

**Figure 6.** JhI-21 functions upstream of serotonin to modulate protein preference. (**A**) Time-dependent protein preference of flies with mutation in one of the *Drosophila* SLC7A5 proteins, JhI-21. Mutation in *JhI-21*, abolished protein preference. The choice was given as 1% sucrose vs 1% sucrose+ 2% autolyzed yeast. Cumulative preference index for Canton S files was 0.6 ± 0.08 whereas *JhI-21* mutant flies was −0.02 ± 0.1 (Student's t-test; P≤0.001). (**B**) Serotonin abundance in the heads of *JhI-21* mutant flies after specified diet treatments. There was no change in serotonin abundance following 24 hr starvation or 3 hr of autolyzed yeast refeeding after starvation. Individual symbols represent measures based on 10 female fly heads, and lines denote the mean (one-way ANOVA; N=5 biological replicates/treatment). (**C**) *JhI-21* mutant flies increase reproductive output normally as concentration of dietary protein increases. (**D**) *JhI-21* mutant flies consume the same amount of sucrose and yeast as control flies regardless of the diet environment (two-way ANOVA). (**E**) Survivorship of *JhI-21* mutants aged on the choice diet. Mutants live significantly longer in these conditions compared with the control flies (log-rank test).
*Figure 6 continued on next page*

*Figure 6 continued*

The following figure supplement is available for figure 6:

**Figure supplement 1.** A role of TOR signaling and *JhI-21* in protein preference and diet-dependent lifespan.

exhibited a normal increase in egg production with dietary protein concentration (*Figure 6C*), and they showed normal feeding in both fixed and choice environments (*Figure 6D*). Nevertheless, mutation in *JhI-21* recapitulated the extended lifespan pattern observed in *Trh* and *5HT2a* mutant flies with a modest (13%) increase in mean lifespan in a fixed-diet condition (*Figure 6—figure supplement 1E*) and a greater (32%) mean lifespan extension in a choice-diet environment (*Figure 6E*). These data suggest that the *JhI-21* amino acid transporter, and possibly SLC7 protein family members in general, are conserved regulators of protein-dependent behavior and physiology and that they may function together with serotonin signaling to modulate aging independently of mechanisms that regulate reproduction and total food intake.

## Discussion

We used a newly designed continuous feeding monitor (FLIC) and a novel dietary paradigm to show that in the fruit fly, *Drosophila melanogaster*, serotonin signaling is part of a reward circuit that is important during meal choice for assessing the value of ingested protein and inducing changes in behavior and aging. We used genetic tools that are specific to serotonergic synthesis in the CNS (*Trh-GAL4*; *Daubert et al., 2010*) and loss-of-function mutants for *Trh* itself (*Neckameyer et al., 2007*) and *5HT2a* (*Nichols, 2007*) to reveal significant spatial specificity of serotonin signaling in the brain for these effects. Based on immunostaining of adult *Drosophila*, Trh is found in 83 neurons per hemisphere of the brain (*Bao et al., 2010*), and 5HT2a is concentrated in small groups of glomeruli such as the ellipsoid body, large field R-neurons, antennal lobe, and a subset of gustatory neurons in the suboesophageal ganglion (*Nichols, 2007*). Moreover, by combining temporal neuronal manipulations with continuous feeding measures we were able to provide, for the first time, evidence that serotonin is acting acutely during the post-ingestive phase to drive behavioral and health outcomes. Mammalian studies have implicated some component of serotonin signaling in feeding regulation and nutrient balance, but these studies have been limited by the lack of spatial and temporal resolution inherent in pharmacological treatments and the technical difficulties of manipulating individual neurons or serotonin receptors. (*Anonymous, 1992*; *Johnston, 1992*; *LeBlanc and Thibault, 2003*; *Leibowitz et al., 1993*; *Leibowitz and Alexander, 1998*; *Leibowitz et al., 1989*; *Magalhães et al., 2010*; *Vargas et al., 2010*).

Our results implicate nutrient valuation as another component of a central homeostatic system that perceives and evaluations nutrient availability to drive behavior and aging. Nutrient homeostasis includes a dynamic process of context-dependent valuation of nutrients, which entails an integration of sensory perception of ecological availability and an internal assessment of nutrient demand. Among such cell non-autonomous mechanisms, sensory perception of nutrients, independent of feeding, is known to modulate aging in several species (*Libert et al., 2007*; *Smith et al., 2008*). This hypothesis begs the questions of whether and how sensory perception influences general serotonin signaling and the specific behavioral and aging effects that we describe. Our discoveries that the SLC7A5 homolog, JhI-21, is required for normal serotonin release following protein feeding and that loss of function in this gene increases lifespan suggest that peripherally expressed amino acid transporters play important roles in a putative central nutrient homeostatic system. In mice, a lysosomal amino acid transporter has been shown to communicate luminal amino acid levels to the canonical cellular amino acid sensor, TOR (*Attardo et al., 2006*; *Wang et al., 2015*). In *Drosophila*, an insect glycine transporter, *Slimfast*, together with certain solute carrier 6 family members are critical for organismal growth (*Colombani et al., 2003*; *Goberdhan et al., 2005*), and several orphan *Drosophila* amino acid transporters in adipocytes regulate ovarian stem cell number (*Armstrong et al., 2014*). In light of these studies, it is compelling to consider a potential amino acid sensing function of JhI-21 and to investigate its temporal and special expression. This may serve as a model to

understand how information on nutrient availability and demand is relayed from key peripheral tissues to the CNS to stimulate appropriate behavioral and physiological responses.

While mutations in *Trh* or *5HT2a* lead to altered dietary choice following short-term starvation, we find no evidence that they influence long-term, *ad libitum* feeding patterns in which flies are long-lived. We believe that these seemingly contradictory results instead indicate complex dynamics of nutrient demand and reward that may be occurring continuously in natural conditions. Despite plentiful food, it is likely that at any given time individual flies comprise a spectrum of protein demand states based on their recent history of feeding. Indeed, there is some indication of this in our long-term FLIC data (*Figure 1A*), which suggest the possibility of a cycling between protein and sugar preference throughout a 24 hr period. This is what might be expected were flies to be switching between foods to balance intake of protein and carbohydrate over time. Our mild starvation treatment would be expected to both enhance protein demand and synchronize this demand among flies, leading to a sensitized assay for protein preference at the population level. Tracer methods are unable to capture such dynamics and thus make it currently a technical challenge to understand these subtle differences in long-term *ad libitum* feeding patterns under conditions used for aging assays. Nonetheless, it is noteworthy that the beneficial effects of reduced serotonin signaling for lifespan become magnified when animals are asked to actively balance their nutrients in choice-diet conditions.

The choice diet presents an interesting environment in which to study mechanisms of aging in *Drosophila.* That the decision-making processes involved in nutrient balance have long-term effects on health and aging is, perhaps, not surprising when considering that such decisions reach beyond immediate fulfillment of nutritional demand. Studies of choice behavior have revealed interesting insights into the effect of dietary self-selection and different life history traits (*Jensen et al., 2015*; *Maklakov et al., 2008*). However, incorporating such natural behaviors into aging studies is rare, and our work provides the first mechanistic insight into their influence. In our experiments, we presented the two macronutrients (carbohydrate and protein) as a choice, expecting this to ameliorate a situation where the animals are forced to ingest excess of a certain nutrient to obtain minimal levels of another (*Simpson and Raubenheimer, 2007*) or where the animals prematurely cease feeding to prevent such excess ingestion at the expense of remaining deficient. Interestingly, young flies appear healthier in the choice diet, with greater protein and TAG abundances and more robust behavioral abilities, including negative geotaxis. These functions deteriorate more rapidly in the choice diet, however, which is consistent with more rapid aging. While the subtleties of the choice environment remain to be elucidated, it is remarkable to note that not only nutrient availability but also nutrient presentation is important, independent of how much of each are consumed. If one accepts this premise, then it is reasonable to consider this environment as a more appropriate husbandry paradigm that better mimics conditions in nature or in human societies where individuals often seek out specific nutrients to satisfy short-term cravings.

Finally, our findings suggest that, even in simple organisms, the brain continuously evaluates key biological states, including nutrient demand and reward, and actively employs simple decision-making processes to affect behavior and physiology and to modulate survival. This is consistent with reports showing that specific genetic and environmental manipulations, such as dietary restriction (*Good and Tatar, 2001*; *Mair et al., 2003*; *Smith et al., 2008*), insulin signaling (*Giannakou et al., 2007*) and mate availability (*Gendron et al., 2014*; *Maures et al., 2014*) rapidly and reversibly affect mortality rates, often within hours or days. It is also notable that the same neural circuits that evaluate internal and external nutritional status to determine what and when to eat also interact with major hormone axes known to influence aging, such as insulin-like and TGFβ signaling (*Domingos et al., 2011*; *You et al., 2008*). Aging therefore has characteristics that resemble a complex behavior that is acutely malleable, susceptible to sensory influences, and strictly controlled by coordinated sets of neurons.

## Materials and methods

### Fly stocks

The following stocks were obtained from Bloomington Stock center; *Canton S.* (RRID:FlyBase_FBst1000081), *w1118*, *Trh^{c1440}* (*Neckameyer et al., 2007*; RRID:BDSC_10531), *5HT2a^{PL00052}*

(*Nichols, 2007*; RRID:BDSC_19367), *JhI-21*[GE15185] (*Jin et al., 2008*; RRID:BDSC_26889), *CCKLR-17D1*[MB02688] (RRID:BDSC_23482), *lr64a*[MB05283] (RRID:BDSC_24610), *S6KII*[G1845] (RRID:BDSC_32601), P{TRiP.GL00267}attP2 (RRID:BDSC_35355), *Gr64f*[MB12243] (RRID:BDSC_27883), *Gr66a*[Δ83] (RRID:BDSC_35528), *Gr93a*[3] (RRID:BDSC_27592), *DopR1*[attP]. For RNAi-mediated knock-down of *5HT2a* we used P{KK110704}VIE-260B from Vienna Drosophila Resource Center. *UAS-dTrpA1* (*Hamada et al., 2008*), *UAS-shi*[ts1] (*Kitamoto, 2001*; RRID:BDSC_44222), *Trh-GAL4* (*Daubert et al., 2010*), *S6KII*[ign-Δ58-1], *4EBP*[Δ], *ppk28*[Δ], *Chico*, *dFoxO*[Δ94] (RRID:BDSC_42220), *NPFR*[c01896] (RRID:BDSC_10747), Orco[2] (RRID:BDSC_23130), *Gr5a*[Δ5], *Gr32a* were gift from P. Garrity (Brandeis University, Waltham, MA), T. Kitamoto (U of Iowa, Iowa city, IA), B.G. Condron (U of Virginia, Charlottville, VA), R. Jackson (Tufts University, Boston, MA), R.J. Wessells (Wayne State University, Detroit, MI), K. Scott (U of California Berkeley, Berkeley, CA), L. Partridge (U college of London, London, UK), S. Waddell (U of Oxford, Oxford, UK), L. Vosshall (The Rockefeller University, New York city, NY), J. Carlson (Yale University, New Haven, NJ), and H. Amrein (Texas A&M, College Station, TX), respectively. All 5HT receptor mutants (*5HT1a*[MB9812], *5HT1a*[MB9978] (RRID:BDSC_27820), *5HT1b*[MB5181] (RRID:BDSC_24240), *5HT1b*[MB5999], *5HT2b*[MB0650] (RRID:BDSC_40810), *5HT2a*[Mi3299] (RRID:BDSC_37177), *5HT2a*[MI00459-GAL4])were gifts from H. Dierick (Baylor College of Medicine, Houston, TX). G.W. Roman shared *Tdc2-GAL4* (RRID:BDSC_9313) and *Th-GAL4* as gifts. A novel *Ast-C* deletion alleles was created using a FLP-FRT recombination mediated strategy described previously (*Parks et al., 2004*) using d00174 and f00146 (Harvard-Exelixis stock center). *Trh*[c1440], *5HT2a*[PL00052], and *JhI-21*[GE15185] mutants were backcrossed at least 8 generations to *w*[1118] prior to any followed up experiments after the candidate screens.

## Husbandry

All fly stocks were maintained on a standard cornmeal-based larval growth medium and in a controlled environment (25°C, 60% humidity) with a 12 Light: 12 dark cycles. If flies contained temperature-sensitive transgenes, they were reared in 23°C and maintained at this temperature as adults until the experiments. We controlled the developmental larval density by manually aliquoting 32μL of collected eggs into individual bottles containing 25 ml of food. (*Linford et al., 2013*) Following eclosion, mixed sex flies were kept on SY10% medium for 4–10 days until they were used for experiments. Unless otherwise noted, we used mated female flies that were between 5–14 days old for the choice experiments. When starvation was required for the feeding assay, we used 1% agar medium to deprive food but prevent dehydration.

## Food intake and choice measurements using the CAFE assay

We used a modified CAFE assay (*Ja et al., 2007*) to measure food intake or food choice as described previously (*Ro et al., 2014*). All choice experiments using the CAFE assay lasted 3 hr in 25°C, 60% relative humidity, and uniform lighting. The Preference Index (PI) was calculated as ''[(Volume of protein+sugar consumed/fly)-(Volume of sugar consumed/fly)]/ [Total volume of food consumed/fly]''. We used either 45 mg/ml bovine albumin serum (BSA; Fisher Scientific) or autolyzed yeast (Bacto yeast extract, BD) as protein sources and 1% sucrose as sugar in the choice experiments. The candidate screen was done using the CAFE assay and BSA as a protein source. For the screen, we added $10^{-5.5}$ μM of denatonium benzoate (Sigma-Aldrich) in the protein-supplemented food to reduce the frequency of false positives.

## Food Choice measurement using the FLIC assay

Details of the feeding experiments using the Fly Liquid-food Interaction Counter (FLIC) system can be found in (*Ro et al., 2014*) and wikiflic.com. Briefly, we filled one side of the food trough with a 1% sucrose+ protein solution and the other trough with a solution of 1% sucrose-only. To avoid positional bias, we alternated the side of each food type across different *Drosophila* feeding monitors (DFMs). After loading the foods, we introduced an individual fly into each behavioral arena using an aspirator. We began the FLIC monitor software (V. 2.0–2.1) before flies were loaded to ensure that no feeding signals were lost during the fly-loading time. Unless otherwise noted, choice experiments lasted 3 hr. A Cumulative PI of an individual was calculated as ''[(Total feeding interaction time on protein+sucrose) - (Total feeding interaction time on sucrose)]/Total feeding interaction time''. We computed a time-dependent feeding PI from an individual fly by calculating fixed feeding PI within a

30 min time-window every 10 min. Flies that did not generate any feeding signals in a given 30 min window were treated as missing data for that period. Average time-dependent PI values and their SEM across biological replicates were therefore calculated based only on flies that exhibited at least one feeding event during the period in question. Initial characterization of the protein feeding behavior when starved and fully-fed were determined by the FLIC system as well as all subsequent follow-up experiments after the candidate screen.

Feeding bout preference index (PI) was used to calculate the PI over a defined number of feeding bouts (contrary to a defined time interval as reflected in the time-dependent PI). This measure permits comparing the food preference within the same number of feeding bouts across different genotypes or treatment groups regardless of when the feeding occurred, hence removing variability in feeding times across individuals (e.g., have removing differences between 'slow-eaters' and 'fast-eaters'). We calculated the Feeding bout PI as' [(Feeding interaction time on protein+sucrose during x number of feeding bouts) - (Feeding interaction time on sucrose during x number of feeding bouts)]/ Total feeding interaction time during x number of feeding bouts". Feeding bout PI were calculated for every subsequent 5 or 10 feeding bouts as noted in the relevant figures.

## Total food intake measurement using tracer dye

Fifteen day old, age-matched female flies were kept on SY10% food, then transferred to test foods mixed with 0.5% blue dye (FD&C Blue no. 1; Spectrum Chemical). We let flies feed on dyed food between 4pm till 1 am the next day (8 hr), capturing one period of 'light-off' when flies normally consuming more than half of their daily food intake (*Ro et al., 2014*). We then froze flies at −20℃ to stop the feeding experiment and homogenized flies in 150 μL PBS + 0.1% Triton X-100 (IBI Scientific). Homogenates were centrifuged at 3,750 × g for 10 min to settle debris, filtered through a 0.4 μm filter, and read absorbance at 630 λ, with 670 λ as a reference wavelength. Absorbance values from 670 λ readings were subtracted from 630 λ readings to correct for background from fly homogenate. For measuring flies' proportional food intake within a choice environment, we had three diet groups where we dyed either 10% sucrose only, or 10% yeast only, or both. We used 10 female flies per sample and 8 biological replicates per genotype and treatment group.

## Quantification of protein, fat, and carbohydrate from fly homogenates

For quantifying total protein, fat, and carbohydrates from fly homogenates, we froze flies after experimental treatment at −20℃, then homogenized in groups of five in 500 μL PBS + 0.1% Triton X-100 (IBI Scientific). Samples were centrifuged at 3,750 × g for 1 min to settle debris. All measurements were based on colorimetric assays that were carried out using a Synergy2 plate reader (BioTek). For triacylglyceride (TAG) measurement, 5 μL homogenate was mixed with 150 μL of 37℃ Infinity Triglycerides reagent (Thermo Scientific) and incubated at 37℃ for 10 min. Absorbance was measured at 520 λ. A serial dilution of 2 mg/mL glycerol was used as a standard. For glucose+trehelose measurement, we mixed 10 μL of homogenate with 100 μL of 37℃ Infinity Glucose reagent (Thermo Scientific), followed by 30 min incubation at 37℃. Absorbance was measured at 340 λ. We created a calibration curve from 2 mg/mL glucose standard. For glycogen measurement, we followed the same protocol as the glucose+trehelose measurement, except that we used 10 μL of homogenates that were treated with 0.5 μL of amyloglucosidase (0.1U/μL), an enzyme that breaks glycogen down to glucose, for 30 min at 37℃. We then subtracted total free-glucose concentrations that were obtained from the initial glucose measurements to compute concentrations of glycogen. For protein measurement, 2 μL of fly homogenate was incubated with 200 μL of (1:50) 4% (w/v) cupric sulfate/BCA Solution (Novagen) at room temperature for 30 min. Absorbance was measured at 562λ. A serial dilution of 2 mg/mL BSA standard were used to construct the calibration curve. For assessing nutrient stores in fully-fed or starved flies we used 10 female files per sample and 10 biological replicates. For assessing metabolites in flies between fixed and choice-diet conditions, we aged 8-day-old female flies in respective diet conditions for 8 days (6 biological replicates).

## Quantification of serotonin using UPLC-MS

### Sample preparation for the UPLC-MS

To extract serotonin and its related metabolites, female flies were snap frozen in liquid nitrogen and then vigorously vortexted to remove heads. The heads were then homogenized with ice-cold 3x volume of acetonitrile (we assumed a single head is equal to 1 µL) using a pestle grinder. We centrifuged the homogenates at 18,000 x g for 5 min and collected the organic phase as a tissue extract. To derivatize our samples prior to the UPLC-MS analysis, 12 µL of each tissue extract sample were benzoylated by the sequential addition of 6 µL of carbonate buffer (sodium carbonate, 100 mM), 6 µL of benzoyl chloride (2% in acetonitrile, v/v), and 6 µL of an internal standard solution (1% $H_2SO_4$ in 20% acetonitrile, v/v) (*Song et al., 2012*). Internal standards contained analytes that had been labeled with $C^{13}$-benzoyl chloride.

### Liquid chromatography

To separate analytes, we used a Waters nanoAcquity UPLC system fitted with an Acquity HSS T3 C18 column (1 × 100 mm, 1.8 µm, 100 Å pore size). Mobile phase A was 10 mM ammonium formate with 0.15% (v/v) formic acid in water. Mobile phase B was acetonitrile. The gradient used was: initial, 0% B; 0.01 min, 17% B; 0.5 min, 17% B; 3 min, 25% B; 3.3 min, 56% B; 4.99 min, 70% B; 5 min, 100% B; 6.5 min; 100% B; 6.51 min, 0% B; 8.5 min, 0% B. We used a total flow rate of 100 µL/min and 5 µL of sample injection in partial loop injection mode. The auto-sampler was kept at ambient temperature whereas the column was held at 27°C.

### Mass spectrometry

An Agilent 6410 triple quadrupole mass spectrometer was used for analyte detection. We used electrospray ionization in positive mode at 4 kV. The gas temperature was 350°C, gas flow was 11 L/min, and the nebulizer was at 15 psi. Benzoylated 5HT was detected by tandem mass spectrometry at a precursor mass to charge ratio (m/z) of 385 and product m/z of 264 using a fragmenter voltage of 140, collision energy of 20 V, and accelerator voltage of 4 CAV. $C^{13}$-labeled internal standard was detected the same except percusor m/z was 397 and product m/z was 270. After the MS analysis, we performed automated peak integration Agilent MassHunter Workstation Quantitative Analysis for QQQ, version B.05.00. All peaks were visually inspected to ensure proper integration. We used 0.5–100 nM synthetic serotonin (5HT; Sigma) diluted in water as a standard to construct a calibration curve. The standard curve was prepared based on the peak area ratio of the standard to the internal standard by linear regression. Below is the detail setting and retention time (RT) for analytes.

## RNA extraction and quantitative PCR

For serotonin receptor mRNA expression analysis, the heads of flies were removed following diet treatments and then frozen at −80°C. We then extracted RNA using TRIzol reagent (Invitrogen) following the manufacturer's protocol. We diluted extracted RNA samples with RNase-free water to an equal concentration and then performed RT-PCR using SuperScript III First Strand cDNA Synthesis (Invitrogen) to generate cDNA. Real-time PCR analysis used Power SYBR Green PCR Master Mix and a StepOne Plus Real-time PCR system (Applied Biosystems). We pulled heads from 50 females per sample and had 3 biological replicates per treatment group.

We used following primers:

```
5HT1a_F: AATAATCAGCCGGACGGAGG
5HT1a_R: GGTGTTGACCGTGTTCGTTG

5HT1b_F: CAGCGATGCGGATGATTA
5HT1b_R: CGAGGCTATCAGATGGTGCT
5HT2a_F: GGCTCGAGGCATCGATCTAC
5HT2a_R: ACGCATATGTTAGGCTCGGG

5HT2b_F: ACTCCAAGAATCACGCCTCG
5HT2b_R: TCGGACGGTCAGGCAATATG
```

```
5HT7_F: TTTTGTGCGACACTTGCCAC
5HT7_R: TTCAGCGCGTTTACTGGGT

RP49_F: ACTCAATGGATACTGCCAG
RP49_R: CAAGGTGTCCCACTAATGCAT
```

## Negative geotaxis assay

Adult female *Canton S.* female flies prepared and collected as previously described (*Linford et al., 2013*) and then subsequently maintained on either on fixed or choice diets for 10 days. Flies were then transferred to the negative geotaxis apparatus using brief $CO_2$ anesthesia, after which they were allowed to recover for 30 min. We used an automated process in which flies were automatically dropped from 24". After freefall, which effectively knocked the flies to the bottom of the chamber, a video camera was triggered and individual fly movements were tracked for 10 sec using the DDrop software developed in our laboratory. From the tracking data we were able to calculate, for each female, the total distance traveled as well as the time required to reach the top of the chamber. We present the time (in seconds) required to reach the top of the chamber, which is highly correlated with total distance traveled.

## Survival analysis

Flies were prepared for survival experiments as previously described (*Linford et al., 2013*), with a slight modification. Briefly, 2–3 day old adult female flies were transferred onto test food medium. For each genotype per treatment, we put 25 flies per vial with 8–10 vial replicates. Flies were transferred to new food three times per week at which time survival was recorded and dead flies removed. For the protein-restriction experiment, we used a fixed mixture of 5% sucrose + 5% yeast as the protein-restricted diet, and a fixed mixture of 5% sucrose +15% yeast as the high protein diet. For the lifespan experiments comparing 'Fixed food' vs 'Choice food' environments, we created inserts that fit into individual vials. These inserts allowed us to expose the flies to two separate sources of food simultaneously. For these experiments we either loaded the same foods on both sides (no choice diets) or a different food on each side (choice diet). Specifically, the no choice diets contained either 10% sucrose in each well, 10% yeast in each well, or a fixed mixture of 10% yeast and 10% sucrose in each well. The choice diet entailed 10% sucrose on one side of the insert and 10% yeast on the other side of the insert. To test effects of a sugar-rich diet on longevity, we provided flies a fixed mixture of 30% sucrose + 5% yeast as the sugar-rich food, or a fixed mixture of 5% sucrose + 5% yeast as the sugar-restricted food.

## Egg laying assay

Seven day old female and male flies that were kept on SY10% food since eclosion and were transferred to five different egg laying media: 10% sucrose mixed with either 1%, 2.5%, 5%, 10%, or 20% yeast. We provided fresh medium every 24 hr for 4 days and counted the number of eggs laid each day. We reported the eggs laid on 4th day when control flies' reproductive output was fully equilibrated to the concentration of dietary protein. We measured 8 biological replicate per genotype and treatment group.

## Statistics

For Cumulative PIs between two genotypes, we used Student's t-test. For comparison involving food preference, gene expression, metabolite amount, serotonin amount, and food consumption with more than two genotypes or treatment groups, we performed one-way ANOVA followed by post-hoc significance test. We took linear regression approach to model the relationship between time spent feeding on either protein or sugar as dependent variable of total feeding time. To test the effects of diet and genotype in flies' food intake of the fixed vs choice diets, we used Two-way ANOVA. Unless otherwise indicated, pairwise comparisons between different treatment survivorship curves were carried out using the statistical package R with DLife, a survival analysis package developed in the Pletcher Laboratory (*Linford et al., 2013*). P-values for survivorship comparisons were

obtained using log-rank test. For testing interaction between genotypes and diets, we used cox-regression analysis to report P-value for the interaction term. In all cases, two-tailed P-values are reported. We also calculated P-values for aging experiments using a mixed effects modeling approach where standard linear models were applied using survival time as an outcome and vial within genotype as a random effect. In all cases, the P-values for the genotype effect on mean life-span was significantly lower than we obtained using non-parametric log-rank statistics. Therefore, for all analyses, we report the P-values from the log-rank analyses because they are more conservative. To test for a diet effect on negative geotaxis, we used Mann-Whitney U-test.

## Acknowledgements

We thank NJ Linford and BY Chung for their critical comments on the manuscript. We also thank the members of the Pletcher Laboratory for their support and comments about experimental design throughout the study. The funders had no role in study design, data collection and analysis, decision to publish, or preparation of the manuscript. Opinions expressed are those of the authors and not necessarily the funders or any other organization.

## Additional information

### Funding

| Funder | Grant reference number | Author |
|---|---|---|
| American Federation for Aging Research | | Jennifer Ro |
| National Institutes of Health | T31 GM007315 | Jennifer Ro |
| National Institutes of Health | T32AG000114 | Jennifer Ro<br>Scott D Pletcher |
| National Institutes of Health | F31AG04769 | Jennifer Ro |
| National Institutes of Health | R31 DK046960 | Robert T Kennedy<br>Paige A Malec |
| National Institutes of Health | R01AG043972 | David B Allison<br>Scott D Pletcher |
| National Institutes of Health | R01GM102279 | Scott D Pletcher |
| National Institutes of Health | R01AG023166 | Scott D Pletcher |
| National Institutes of Health | R01AG030593 | Scott D Pletcher |
| Ellison Medical Foundation | | Scott D Pletcher |

The funders had no role in study design, data collection and interpretation, or the decision to submit the work for publication.

### Author contributions

JR, Conception and design, Acquisition of data, Analysis and interpretation of data, Drafting or revising the article; GP, PAM, YL, Acquisition of data, Analysis and interpretation of data; DBA, RTK, Drafting or revising the article, Contributed unpublished essential data or reagents; SDP, Conception and design, Analysis and interpretation of data, Drafting or revising the article

### Author ORCIDs

Scott D Pletcher, http://orcid.org/0000-0002-4812-3785

## Additional files

### Supplementary files

• Supplementary file 1. A list of candidates used in the reverse genetic screen of protein preference. The list is showing the genotype or treatment of candidates in descending orders of average preference index (PI). The summary of PI and candidate is graphically depicted in *Figure 2A*.

• Supplementary file 2. Summary statistics of samples represented in Time-dependent preference index (PI) plots. The list is showing genotype, cumulative PI, standard error of mean (SEM), and sample size (N) of the flies used in figures as noted.

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
