## [Decision Letter]

Thank you for submitting your article "Serotonin signaling mediates protein valuation and aging" for consideration by *eLife*. Your article has been reviewed by three peer reviewers, one of whom, Andrew Dillin, is a member of our Board of Reviewing Editors, and the evaluation has been overseen by a Senior Editor. The following individuals involved in the review of your submission have agreed to reveal their identity: Andrew Dillin (Reviewing Editor and Reviewer #1), William Mair (Reviewer #2) and Michael Petrascheck (Reviewer #3).

The reviewers have discussed the reviews with one another and the Reviewing Editor has drafted this decision to help you prepare a revised submission.

In this body of work the authors investigate whether there is an intersection between food choice (sugar vs. protein) and aging. Using a novel assay developed in the lab that monitors contact of flies with food they can accurately measure how long and how much a fly "chooses" a certain type of food. Using this assay, after a brief starvation, the group finds that protein rich diets are preferred. Asking how this is perceived, the group goes through a battery of sensory-type mutants and find that serotonin synthesis is essential for this process. They go on to find that only 1 of the 5 serotonin receptors is required (which is very, very interesting) and can recreate the mutant phenotypes with drug inhibitors. Although not entirely clear to the non-fly person, they find that a juvenile hormone receptor mutant also blocks this response and appears to act upstream of serotonin (at least transcriptionally). The group goes onto to ask the temporal ordering of serotonin in the response and also find a key link to longevity.

In summary, the work is very well presented and extremely novel. However, all three reviewers feel that the introduction to the juvenile hormone part needs more background and reasoning. All three reviewers are concerned about sample size and given that this is an automated system, more analysis should be performed to bring up the n.

Reviewer #1:

In this body of work the authors whether there is an intersection between food choice (sugar vs. protein) and aging. Using a novel assay developed in the lab that monitors contact of flies with food they can accurately measure how long and how much a fly "chooses" a certain type of food. Using this assay, after a brief starvation, the group finds that protein rich diets are preferred. Asking how this is perceived, the group goes through a battery of sensory-type mutants and find that serotonin synthesis is essential for this process. They go on to find that only 1 of the 5 serotonin receptors is required (which is very, very interesting) and can recreate the mutant phenotypes with drug inhibitors. Although not entirely clear to the non-fly person, they find that a juvenile hormone receptor mutant also blocks this response and appears to act upstream of serotonin (at least transcriptionally). The group goes onto to ask the temporal ordering of serotonin in the response and also find a key link to longevity.

In summary, the work is very well presented and extremely novel. I do not think further experiments will improve the paper and its finding and fully support publication.

Reviewer #2:

The author outline their interesting findings on the links between dietary composition and lifespan in fruit flies via the effects on reward of food rather than nutritional value per se. They show short term starvation induces specific food preferences upon re-feeding using their elegant FLIC food choice assay. Fed flies prefer sugar only food where as previously starved flies show preference for a sugar/protein mix independent of calorific content. They performed a candidate genetic screen to identify the underlying mechanism and defined a clear role for serotonin signaling as flies lacking *Trh* showed reduced protein preference, and this effect is specifically while the food choice is being made and not prior. Interestingly, hyper activating serotonin signaling drove preference for protein containing food suggesting the value of protein was increased rather than serotonin mimics protein sensing. *Trh* mutants themselves are long lived but most intriguingly, flies given nutritional choice in their diet live very short compared to those on a fixed ratio media. Strikingly however, mutants in *Trh* or 5HT2 lived 90% longer in these conditions despite eating as much. Finally, mutation to the animal acid transporter *jhI-21* increases lifespan and dampens serotonin signaling in response to protein. Experiments are well thought out and executed and overall I think it’s a worthy and interesting contribution to *eLife*.

The key point of this paper is that protein valuation/reward itself ages flies which is very novel and fits well with data from this lab previously on the effects of food odor as a suppressor of dietary restriction. However, in fixed food ratios, increasing protein suppresses lifespan in *Trh* mutants despite their lack of protein 'reward'. The authors explain away this effect by the concomitant changes in sugar intake. The manuscript would benefit from further discussion on this point and ideally some experimental support – i.e., if one were to increase protein in a fixed diet while decreasing sugar (i.e., removing its toxicity) would the authors predict that wild type animals would show decreased lifespan due to increased serotonin reward while now *trh* mutant would be refractory?

An additional concern is how well the *jhI-21* data integrate with the rest. The data would predict that the effects of *jhI-21* loss would be repressed by the activated serotonin signaling lines.

Reviewer #3:

In the provided paper the authors present the very interesting finding that protein value of a meal is mediated by serotonergic signalling via the receptors 5HT2a. The authors show that starvation induces flies to prefer protein rich food over sugar rich food and that this preference is mediated by serotonerigic signaling, requires tryptophan hydroxylase, the 5HT2a receptor and the solute carrier JhI-21. Temporal control over serotonin signaling using the shibiri system establishes that the serotonergic signal to be active during the re-feeding stage after starvation. This is further confirmed by qPCR or for the 5HT2a receptor. Interestingly, animals that are allowed to choose their food components show a shorter lifespan but only in animals with intact serotonin signaling. Animals with reduced serotonin signaling live dramatically longer in the choice paradigm even though the serotonin mutants also have a higher protein and triacylclycerol level and are not calorically restricted. Overall, an interesting and convincing study.

Behavior experiments generally result in noisy readouts. The current graphs provide limited insights into how consistent the behaviors are.

Please address the following points for the behavior graphs by adding to the actual graph or by providing an second supplementary version:

Some kind of a confidence interval/standard deviation in Figure 1, Figure 2 are necessary to judge how robust the behavior is.

When calculating the PI, are there genotypes/interventions that cause more "missing" flies than others? In 2B Trh seems to very robust, all circles about the same size suggesting that there are hardly any "missing" flies that did not produce a feeding signal during the 3 h. For the 5TH2A in 2D it seems that most flies don't eat between 60 and 120 min, suggesting that most of the negative PI is caused by very few flies.

Do some mutants show differences in mobility that could influence the preference index?

---

## [Author Response]

*Reviewer #2:*

*The author outline their interesting findings on the links between dietary composition and lifespan in fruit flies via the effects on reward of food rather than nutritional value per se. They show short term starvation induces specific food preferences upon re-feeding using their elegant FLIC food choice assay. Fed flies prefer sugar only food where as previously starved flies show preference for a sugar/protein mix independent of calorific content. They performed a candidate genetic screen to identify the underlying mechanism and defined a clear role for serotonin signaling as flies lacking Trh showed reduced protein preference, and this effect is specifically while the food choice is being made and not prior. Interestingly, hyper activating serotonin signaling drove preference for protein containing food suggesting the value of protein was increased rather than serotonin mimics protein sensing. Trh mutants themselves are long lived but most intriguingly, flies given nutritional choice in their diet live very short compared to those on a fixed ratio media. Strikingly however, mutants in Trh or 5HT2 lived 90% longer in these conditions despite eating as much. Finally, mutation to the animal acid transporter jhI-21 increases lifespan and dampens serotonin signaling in response to protein. Experiments are well thought out and executed and overall I think it’s a worthy and interesting contribution to eLife.*

*The key point of this paper is that protein valuation/reward itself ages flies which is very novel and fits well with data from this lab previously on the effects of food odor as a suppressor of dietary restriction. However – in fixed food ratios increasing protein suppresses lifespan in Trh mutants despite their lack of protein 'reward'. The authors explain away this effect by the concomitant changes in sugar intake. The manuscript would benefit from further discussion on this point and ideally some experimental support – i.e. if one were to increase protein in a fixed diet while decreasing sugar (i.e. removing its toxicity) would the authors predict that wild type animals would show decreased lifespan due to increased serotonin reward while now trh mutant would be refractory?*

The reduced lifespan of *Trh* (and to a lesser extent 5HT2a) mutants on a high-protein fixed-diet was surprising to us, and (as described in the manuscript) it stimulated our investigation of the choice diet environment. Indeed, because we found the dramatic effects of a choice diet equally interesting, we have been pursuing experiments that are designed to understand how and why the choice diet produces the effects it does. While the details of this work are not pertinent, it has revealed that flies carrying a null mutation in 5HT2a are, for all practical purposes, immune to changes in lifespan, physiology, and health brought about by protein manipulation in the diet. We currently believe that the difference between *Trh* and 5HT2a mutants in these studies is due to the nature of the mutant alleles. The *Trh* allele is a hypomorph (complete loss of *Trh* is lethal during development) that retains some serotonergic signaling. The molecular details concerning how serotonin impacts lifespan through 5HT2a in response to protein reward are currently under investigation and, we believe, outside the scope of this current manuscript.

Finally, we have, in effect, carried out the experiment suggested by this reviewer. Figure 4 (yeast only panel) shows the lifespan in the extreme case, no added sugar, only yeast. In this case, while *Trh* mutants are still long-lived, the effect is much reduced compared to the traditional fixed-diet recipe.

*An additional concern is how well the jhI-21 data integrate with the rest. The data would predict that the effects of jhI-21 loss would be repressed by the activated serotonin signaling lines.*

We have modified the text to clarify this transition and (recognizing comments from other reviewers as well) to integrate this section better into the manuscript as a whole.

*Reviewer #3:*

*In the provided paper the authors present the very interesting finding that protein value of a meal is mediated by serotonergic signalling via the receptors 5HT2a. The authors show that starvation induces flies to prefer protein rich food over sugar rich food and that this preference is mediated by serotonerigic signaling, requires tryptophan hydroxylase, the 5HT2a receptor and the solute carrier JhI-21. Temporal control over serotonin signaling using the shibiri system establishes that the serotonergic signal to be active during the re-feeding stage after starvation. This is further confirmed by qPCR or for the 5HT2a receptor. Interestingly, animals that are allowed to choose their food components show a shorter lifespan but only in animals with intact serotonin signaling. Animals with reduced serotonin signaling live dramatically longer in the choice paradigm even though the serotonin mutants also have a higher protein and triacylclycerol level and are not calorically restricted. Overall an interesting and convincing study.*

*Behavior experiments generally result in noisy readouts. The current graphs provide limited insights into how consistent the behaviors are.*

*Please address the following points for the behavior graphs by adding to the actual graph or by providing an second supplementary version:*

*Some kind of a confidence interval/standard deviation in Figure 1, Figure 2 are necessary to judge how robust the behavior is.*

*When calculating the PI, are there genotypes/interventions that cause more "missing" flies than others? In 2B Trh seems to very robust, all circles about the same size suggesting that there are hardly any "missing" flies that did not produce a feeding signal during the 3 h. For the 5TH2A in 2D it seems that most flies don't eat between 60 and 120 min, suggesting that most of the negative PI is caused by very few flies.*

We recognize the difficulty in evaluating individual variability in these types of measures and appreciate the concern presented by this reviewer. Indeed, deriving new statistical analyses and error representation for these types of continuous data in which different flies (or none at all) contribute to the observable measure of the group turns out to be a significant statistical problem, which is, to our knowledge, unsolved at this point. Indeed, no other methods for measuring feeding behavior provide the ability to quantitatively assess individual variability (nearly all derive from group statistics). Ongoing theoretical research is being carried out to solve this problem. Nevertheless, we have addressed this concern by: (i) including manageable cumulative statistics, together with relevant error measures, (ii) adding new data from replicate experiments that establish robustness, and (iii) adding new analyses and data representations that, while in some ways unorthodox, provide additional support for our inference.

We have included statistical details that represent the final PI values and their relevant significant differences among treatments, which reflect variation among individuals. We considered adding these values as insets to the plots themselves, but ultimately decided against this because they became too “busy.”

Understanding the reviewers’ desire to make these values more accessible to the reader, we created a new supplementary table with this information ([Supplementary-material SD2-data] in the revision). As we mentioned above, providing meaningful confidence intervals on individual time points in the time-dependent PI plots is theoretically difficult not only because we have repeated measures but also because different flies contribute to different points. It is currently unclear how to take into account this correlation structure when representing error.

To illustrate robustness and reproducibility, we have added new replicate and representative results from food preference assays that repeat and extend those in Figure 2. The first set replicates results using the *Trh* mutation (Figure 2—figure supplement 1). To complement the main text, which reported results from the PL00052 5HT2a allele, and the supplementary data, which reported results using the MI3299 5HT2a allele, we have added one more replicate and representative experiments using a third 5HT2a mutant genotype: flies trans-heterozygous for the two loss of function 5HT2a mutant alleles with autolyzed yeast (Figure 2—figure supplement 1). We have also added a new replicate of *JhI-21* mutant preference (Figure 6—figure supplement 1).

Regarding the “missing flies issue” noted by the reviewer, we would point out that all of the flies in each experiment provide some amount of preference data. In other words, all flies eat one food or another at several times throughout the experiment, and therefore each fly provides a cumulative PI value that we now represent in the revision (see comment above). However, they obviously do not all feed at the same time. We tried to be as transparent as possible about this by representing the time-dependent PI plots with points that are proportional to the number of flies contributing to them. There is, however, a second way of representing feeding data, which is not by time but is instead by feeding bout. In other words, rather than asking what the preference index is for those flies that are feeding between 30 and 60min, we could ask what the preference index is for each fly over its first 5 meals (or 10 or 15 meals…) regardless of when those meals occurred. In this way, the PI cannot be influenced by an unusual food choice by a small number flies because every included fly would contribute the same amount of information to each point (e.g., every fly in the experiment feeds at least five times). While not every fly would exhibit 10 or 15 meals, those that do contribute equally to the point and again error bars are meaningful. These “Feeding bout PI” plots are now presented in the supplementary figures as: Figure 1, Figure 2—figure supplement 2. We present Feeding bout PI only for select bout numbers because, while we could represent it as continuous (i.e., for 5,6,7,8, etc. feeding bouts), the statistics are intractable.

*Do some mutants show differences in mobility that could influence the preference index?*

Differences in mobility are unlikely to affect PI but would manifest in overall consumption or total interactions with food in general. With the exception of TRPA1-mediated activation of TrH^+^ neurons, which significantly decreased intake as noted in the manuscript, we do not observe significant changes in overall food interactions (this is supported by the food intake measures presented in Figure 4).